# Mapping HIV prevalence in Nigeria using small area estimates to develop a targeted HIV intervention strategy

**Caitlin O'BRIEN-CARELLI**[1], **Krista STEUBEN**[1], **Kristen A. STAFFORD**[2,3], **Rukevwe ALIOGO**[4], **Matthias ALAGI**[5], **Casey K. JOHANNS**[1], **Jahun IBRAHIM**[5], **Ray SHIRAISHI**[6], **Akipu EHOCHE**[4], **Stacie GREBY**[5], **Emilio DIRLIKOV**[6], **Dalhatu IBRAHIM**[5], **Megan BRONSON**[6], **Gambo ALIYU**[2]\*, **Sani ALIYU**[7]\*, **Laura DWYER-LINDGREN**[1], **Mahesh SWAMINATHAN**[5☯], **Herbert C. DUBER**[1☯], **Man CHARURAT**[2,3☯]\*

**1** Institute for Health Metrics and Evaluation, University of Washington, Seattle, WA, United States of America, **2** CIHEB, University of Maryland School of Medicine, Baltimore, MD, United States of America, **3** Institute of Human Virology, University of Maryland School of Medicine, MD, United States of America, **4** CIHEB, MGIC–An Affiliate of the University of Maryland Baltimore, Abuja, Nigeria, **5** U.S. Centers for Disease Control and Prevention, Abuja, Nigeria, **6** U.S. Centers for Disease Control and Prevention, Atlanta, GA, United States of America, **7** National Agency for the Control of AIDS, Abuja, Nigeria

☯ These authors contributed equally to this work.
\* mcharurat@ihv.umaryland.edu (MC); galiyu@naca.gov.ng (GA); sanialiyu1@gmail.com (SA)

**Data Availability Statement:** All relevant data are within the paper and its Supporting Information files.

## Abstract

### Objective

Although geographically specific data can help target HIV prevention and treatment strategies, Nigeria relies on national- and state-level estimates for policymaking and intervention planning. We calculated sub-state estimates along the HIV continuum of care in Nigeria.

### Design

Using data from the Nigeria HIV/AIDS Indicator and Impact Survey (NAIIS) (July–December 2018), we conducted a geospatial analysis estimating three key programmatic indicators: prevalence of HIV infection among adults (aged 15–64 years); antiretroviral therapy (ART) coverage among adults living with HIV; and viral load suppression (VLS) rate among adults living with HIV.

### Methods

We used an ensemble modeling method called stacked generalization to analyze available covariates and a geostatistical model to incorporate the output from stacking as well as spatial autocorrelation in the modeled outcomes. Separate models were fitted for each indicator. Finally, we produced raster estimates of each indicator on an approximately 5×5-km grid and estimates at the sub-state/local government area (LGA) and state level.

### Results

Estimates for all three indicators varied both within and between states. While state-level HIV prevalence ranged from 0.3% (95% uncertainty interval [UI]: 0.3%–0.5%]) to 4.3%

**Funding:** This project is supported by the President's Emergency Plan for AIDS Relief (PEPFAR) through the Centers for Disease Control and Prevention (CDC) under the cooperative agreement #U2GGH002108 to the University of Maryland, Baltimore and by the Global Funds to Fight AIDS, Tuberculosis, and Malaria through the National Agency for the Control of AIDS, Nigeria, under the contract # NGA-H-NACA to the University of Maryland, Baltimore. The findings and conclusions in this report are those of the authors and do not necessarily represent the official position of the funding agencies.

**Competing interests:** The authors have declared that no competing interests exist.

(95% UI: 3.7%–4.9%), LGA prevalence ranged from 0.2% (95% UI: 0.1%–0.5%) to 8.5% (95% UI: 5.8%–12.2%). Although the range in ART coverage did not substantially differ at state level (25.6%–76.9%) and LGA level (21.9%–81.9%), the mean absolute difference in ART coverage between LGAs within states was 16.7 percentage points (range, 3.5–38.5 percentage points). States with large differences in ART coverage between LGAs also showed large differences in VLS—regardless of level of effective treatment coverage—indicating that state-level geographic targeting may be insufficient to address coverage gaps.

## Conclusion

Geospatial analysis across the HIV continuum of care can effectively highlight sub-state variation and identify areas that require further attention in order to achieve epidemic control. By generating local estimates, governments, donors, and other implementing partners will be better positioned to conduct targeted interventions and prioritize resource distribution.

## Introduction

In Sub-Saharan Africa (SSA), human immunodeficiency virus (HIV) and the availability of testing and treatment services varies substantially at the national, state, and sub-state levels [1–6]. This variation occurs across the entire continuum of care, including knowledge of HIV status among persons living with HIV (PLHIV), antiretroviral therapy (ART) coverage, and viral load suppression (VLS) among PLHIV receiving ART. Despite this geographic heterogeneity, governments and implementing partners continue to rely on national and state-level estimates for policymaking and intervention planning, which can obscure underlying, local disparities in epidemiology and service provision [3, 5, 7].

Studies suggest that using geographically-specific data for developing targeted HIV strategies can be more efficient and effective than a single uniform policy, especially when combined with targeted interventions for key population groups, such as female sex workers and men who have sex with men [2, 3, 5, 8]. A simulation study from Kenya found that an HIV prevention strategy that considers geographic prioritization could generate a 33% reduction in the total number of new infections over a 15-year period compared to a strategy that uniformly distributes resources [9]. Interventions adapted to the sub-state or district level can be especially effective. Studies in Botswana [10], Kenya [6], Lesotho [11], Malawi [12], Rwanda [13], and South Africa [14] indicate that geographic targeting could increase the impact of prevention interventions, with some studies identifying "hotspots" for greater prioritization in geographic areas as small as the community level [15]. Despite the recent evidence that shows the usefulness of geospatial estimates for targeted intervention planning and policymaking [16, 17], there are few examples describing the derivation and practical application of sub-state geospatial estimates for targeted intervention planning.

The 2018 Nigeria HIV/AIDS Indicator and Impact Survey (NAIIS), the largest HIV-focused population-based survey ever conducted in the most populous country in SSA, highlighted important geographic disparities across the continuum of care. NAIIS revealed that Nigeria's national HIV prevalence (1.3% among adults aged 15–49 years) was nearly half of previous estimates based on biannual antenatal care sentinel surveys and the National HIV/AIDS Reproductive Health Survey [18]. In response to this information, the Government of Nigeria and its implementing partners adopted a revised strategic framework that classified

states into "surge" areas for intensified scale-up of HIV case identification and treatment based on the number of PLHIV and ART coverage rates [19]. In addition, these partner organizations developed targeting strategies specific to the 774 local government areas (LGAs), or district-level intervention plans. Here, we present the derivation and subsequent utilization of sub-state estimates of HIV prevalence, ART coverage, and VLS in Nigeria.

## Methods

The 2018 NAIIS (July–December 2018) was a cross-sectional, two-stage nationally representative household survey of 83,909 households from 4,035 enumeration areas selected using a probability proportion to size method across all 36 states and the Federal Capital Territory (FCT) with an equal-size approach to achieve an estimated sample size of 3,700 blood specimens from each state [18]. The survey was designed to assess the prevalence of HIV-related indicators including HIV prevalence, knowledge of HIV status, ART coverage, and VLS, as well as sociodemographic characteristics including age, sex, urban or rural residence, marital status, level of education, and wealth quintile [20]. Written informed consent was obtained for all adult participants and captured electronically. Parental/guardian written consent for minors aged 14 year and under, and written assent from children aged 10–14 years, was captured electronically. Ethical approval was received from the Nigeria Research Ethics Committee, the United States Centers for Disease Control and Prevention, and the University of Maryland, Baltimore Institutional Review Boards.

We conducted a geospatial analysis using 2018 NAIIS data, producing estimates of three HIV-related indicators: prevalence of HIV infection among adults (aged 15–64 years); ART coverage among adults living with HIV; and virus suppression rates among adults living with HIV. We produced raster estimates of each indicator on an approximately 5×5-km grid as well as estimates at the LGA and state level.

### Data

For the purposes of this analysis, we subset the NAIIS individual level data to adults and created three binary individual-level variables: HIV status, ART status, and VLS. HIV status was defined for all adults and was set to 1 for those who tested positive for HIV according to the Nigeria National HIV Testing Guideline [21] using Geenius HIV 1/2 Supplemental Assay as a confirmatory test; HIV status was set to 0 otherwise. ART status was defined for adults who were HIV-positive and was set to 1 if a respondent was aware of their status and reported that they were receiving ART or if their serum assay for antiretroviral medications was positive; ART status was set to 0 otherwise. VLS was defined for adults who were HIV positive and was set to 1 for respondents who were receiving ART (ART status, 1) and had an HIV viral load concentration <1,000 copies/mL and was set to 0 otherwise. This dataset was then merged with the (randomly displaced) [22] cluster-level GPS coordinates.

The NAIIS sample included 206,996 eligible adults; of these, 173,716 gave consent, were interviewed, participated in a blood draw, had non-missing HIV status, and were included in this analysis [18]. For the analysis of HIV prevalence, we aggregated individual-level responses by cluster, calculating both the weighted mean of HIV status and the effective sample size, approximated based on the sample weights using the Kish approximation [1]. We then calculated the effective number of HIV cases in each cluster's sample by multiplying the effective sample size by the weighted HIV prevalence. For the analysis of ART and VLS prevalence among PLHIV age 15–64 years (n = 2,739) with non-missing information for ART and VLS (n = 2,705), we followed the same procedure after sub-setting the dataset to respondents who

were HIV positive. In all cases, the sample weights used were the person-level blood draw weights.

Our analysis also was informed by a suite of raster covariates previously compiled for a geospatial analysis of HIV prevalence in SSA [1]. These covariates included travel time to the nearest settlement of more than 50,000 inhabitants (synoptic); total population (2018); night-time lights (2013); urbanicity (2016); malaria incidence (2017); prevalence of male circumcision (2017); prevalence of self-reported sexually transmitted infection symptoms (2017); prevalence of marriage or living with a partner as married (2017); prevalence of one's current partner living elsewhere (2017); prevalence of condom use at last sexual encounter (2017); prevalence of reporting ever having had intercourse among young adults (2017); and prevalence of multiple sexual partners in the past year (2017). For population estimates, we used gridded populations from WorldPop [23, 24]. Administrative boundaries were obtained from the Office of the Surveyor-General of the Federation of Nigeria. When describing the regional variation, we used common navigational directions in relation to the figures.

## Modeling

The modeling approach used for this analysis was similar to one used for an earlier geospatial analysis of HIV prevalence in SSA [1]. This approach involved two stages. First, we used stacked generalization (stacking) to leverage information from available covariates. Second, we used a geostatistical model to incorporate the output from stacking as well as spatial autocorrelation in the modeled outcomes. Separate models were fitted for each of the three indicators.

Stacking is an ensemble modeling method that combines predictions from multiple different models (sub-models) to capitalize on the strengths of different modeling approaches and increase the overall predictive validity of the resulting estimates compared to utilizing a single model. Our approach follows that described by Bhatt and colleagues [25] and previously applied many other mapping efforts, including for HIV prevalence in SSA [1]. For this analysis, we fitted two sub-models: a generalized additive model and boosted regression trees.

Following stacking, we modeled the prevalence of each indicator using a Bayesian hierarchical logistic regression model that incorporated covariates via the output from the stacking procedure and also explicitly accounted for spatial autocorrelation through a spatially correlated random effect term. For HIV prevalence, we treated the number of HIV-positive adults, $Y_i$, among a sample of size $N_i$ in cluster $i$ as a binomial random variable and modeled the logit-transformed prevalence ($p_i$) as a linear combination of an intercept ($\beta_0$), covariate effects ($\boldsymbol{\beta_1} \cdot \boldsymbol{X_i}$), and a spatially correlated random effect term ($Z_i$):

$$Y_i \sim \text{Binomial}(p_i, \ N_i)$$

$$\text{logit}(p_i) = \beta_0 + \boldsymbol{\beta_1} \cdot \boldsymbol{X_i} + Z_i$$

$$Z_i \sim \text{GP}\left(0, \ \Sigma_{space}\right)$$

The spatially correlated random effect ($Z_i$) was modeled as a Gaussian process with mean 0 and a covariance matrix given by a spatial Matérn covariance function ($\Sigma_{space}$) controlled by two hyper-parameters, the spatial range ($\rho$) and marginal standard deviation ($\sigma$). Penalized complexity priors [5, 6] were assigned for these hyper-parameters. Relatively diffuse mean-0 Gaussian priors were assigned for fixed effects ($\beta_0$ and $\boldsymbol{\beta_1}$).

The models for ART and VLS are similar, with $Y_i$ indicating the number of HIV-positive adults receiving ART or VLS among a sample of HIV-positive adults ($N_i$). On the basis of

cross-validation exercises, we modified the analytic approach for ART and VLS to include only the second stage geostatistical model, which reduces to

$$Y_i \sim \text{Binomial}(p_i, \ N_i)$$

$$\text{logit}(p_i) = \beta_0 + Z_i$$

$$Z_i \sim \text{GP}\left(0, \ \Sigma_{space}\right)$$

Geostatistical models were fitted in R-INLA [7] using the Stochastic Partial Differential Equation approach [8] to approximate the spatial Gaussian random field ($Z_i$). After fitting this model, we generated 1,000 draws from the approximated joint posterior distribution of all model parameters and used these parameter draws to construct 1,000 draws of prevalence ($p_i$) for each 5×5-km grid cell. Prevalence draws were scaled multiplicatively to calibrate the estimates such that the population-weighted mean across all pixels equals the national-level design-based estimates from NAIIS. Final point estimates for each 5×5-km grid cell were calculated from the mean of these 1,000 draws after calibration to the national totals, and 95% uncertainty intervals (UI) were calculated from the 2.5th and 97.5th percentile of these draws.

Each draw was also aggregated within each LGA and state by taking the population-weighted average of prevalence across all grid cells or fractions of grid cells contained within that LGA or state. This resulted in 1,000 draws of prevalence for each LGA and state, and final estimates and UIs were derived from the mean and 2.5th and 97.5th percentile of these draws, respectively. For calibration and aggregation of HIV prevalence, population weighting was undertaken using the WorldPop raster described in the "Data" section. For calibration and aggregation of ART and VLS, we used the estimated number of PLHIV derived from the HIV prevalence model for population weighting.

## Results

Consistent with the primary findings of the NAIIS study [20], we estimated that HIV prevalence in Nigeria among adults aged 15–64 years in 2018 was 1.36% (95% UI: 1.30%–1.44%). National ART coverage was estimated at 45.3% (42.8%–48.0%), and virus suppression rate (or effective ART coverage) was 36.6% (34.8%–38.6%).

### Differences in HIV prevalence, ART coverage, and VLS rates between states

HIV prevalence substantially varied between states, ranging from a minimum of 0.34% (95% UI: 0.25%–0.45%) in Zamfara to 4.3% (95% UI: 3.7%–4.9%) in Akwa Ibom (**Table 1**). The mean HIV prevalence among states was 1.4%, with a difference of 1.8 percentage points between the 10th and 90th percentiles. States with higher HIV prevalence were located in the

**Table 1. Differences in estimated HIV indicators between states\* (n = 37) in Nigeria (2018).**

|  | Min. | Max. | –Difference Max—Min. | Max./Min. | 10th percentile | 90th percentile | 90th - 10th | 90th /10th |
|---|---|---|---|---|---|---|---|---|
| **HIV prevalence (%)** | 0.3 | 4.3 | 3.9 | 12.5 | 0.5 | 2.3 | 1.8 | 4.9 |
| **ART coverage (%)** | 25.6 | 76.9 | 51.4 | 3.0 | 28.7 | 69.0 | 40.3 | 2.4 |
| **VLS (%)** | 18.8 | 66.2 | 47.4 | 3.5 | 21.5 | 56.2 | 34.7 | 2.6 |

\*Includes 36 states and the Federal Capital Territory (FCT)

Abbreviations: ART, antiretroviral therapy; VLS, viral load suppression <1,000 copies/mL.

South and Southeast regions, with the three highest prevalence states—Akwa Ibom (4.3% [95% UI: 3.7%–4.9%]), Benue (3.6% [95% UI: 3.1%–4.2%]), and Rivers (3.2% [95% UI: 2.8%–3.7%])—located in the Southeast region. States in the North and Northwest regions had lower HIV prevalence, with estimated HIV prevalence below 1% observed in most states in the North. Relative differences were also high; estimated HIV prevalence in the highest prevalence state was 12.5 times higher than in the lowest prevalence state, with a 4.9 times difference between the 10th and 90th percentiles.

The percentage of PLHIV receiving ART also varied substantially by state, with an absolute difference between the state with the highest ART coverage (Nasarawa; 76.9% [95% UI: 70.6%–82.2%]) and the lowest ART coverage (Sokoto; 25.6% [95% UI: 13.9%–40.6%]) of 51.4 percentage points. Mean ART coverage among states was 46.5% with a difference of 40.3 percentage points between the 10th and 90th percentiles. Although these absolute differences were greater than the corresponding differences in HIV prevalence due to the magnitude of the estimates, relative differences were smaller.

States with the highest ART coverage were located in Central and Eastern Nigeria, including Nasarawa (76.9% [95% UI: 70.6%–82.2%]), Benue (75.0% [95% UI: 69.7%–79.7%]), Gombe (72.8% [95% UI: 63.5%–80.4%]), Plateau (71.3% [95% UI: 63.2%–78.2%]), and the FCT (67.5% [95% UI: 58.5%–76.0%]). States with the lowest ART coverage were located in the South and Northwest, including Sokoto (25.6% [95% UI: 13.9%–40.6%]), Bayelsa (26.3% [95% UI: 19.5%–34.4%]), Rivers (27.3% [95% UI: 21.7%–33.2%]), and Akwa Ibom (27.9% [95% UI: 23.4%–32.9%]). Akwa Ibom and Rivers, in particular, had HIV prevalence estimates over 3% and ART coverage estimates below 30%, indicating treatment coverage gaps.

The percentage of PLHIV with VLS followed a similar geographic trend; states with low ART coverage had low VLS rates, and states with high ART coverage had high VLS rates. States with the highest VLS rates were located in Central and Eastern Nigeria, whereas states with low VLS rates were located primarily in the South and Northwest. The lowest VLS rates were in Bayelsa (18.8% [95% UI: 13.4%–25.4%]), compared to Nasarawa, which had the highest VLS rate (66.2% [95% UI: 59.6%–72.4%]; Table 1). Mean VLS rate among states was 37.7% with an absolute difference of 34.7 percentage points between the 10th and 90th percentiles. Relative differences were also similar to the percentage of PLHIV on ART: the state with the highest ART coverage had effective coverage 3.5 times higher than the lowest, and the 90th percentile was 2.6 times higher than the 10th.

## Differences in HIV prevalence, ART coverage, and VLS rates between LGAs

HIV prevalence also substantially varied between LGAs. Estimated HIV prevalence ranged from 0.2% (95% UI: 0.1%–0.5%) to 8.5% (95% UI: 5.8%–12.2%; Table 2), and there was an absolute difference of 2.2 percentage points between the 10th and 90th percentiles (Table 2). High-prevalence LGAs were mostly located in high prevalence states, so LGAs with the highest HIV prevalence (>3%) were mostly located in Southern and Eastern Nigeria, whereas low prevalence LGAs were clustered in the North and Northwest regions (Fig 1). Relative

**Table 2. Differences in HIV indicators between local government areas (LGAs; n = 774) in Nigeria (2018).**

|  | Min. | Max. | Max—min. | Max/min. | 10th percentile | 90th percentile | 90th - 10th | 90th /10th |
|---|---|---|---|---|---|---|---|---|
| **HIV prevalence** | 0.2 | 8.5 | 8.3 | 37.7 | 0.4 | 2.6 | 2.2 | 6.4 |
| **ART coverage (%)** | 21.9 | 81.9 | 60.1 | 3.7 | 28.0 | 66.8 | 38.8 | 2.4 |
| **Viral suppression (%)** | 17.8 | 71.0 | 53.2 | 4.0 | 20.5 | 54.8 | 34.3 | 2.7 |

## HIV prevalence by state and LGA

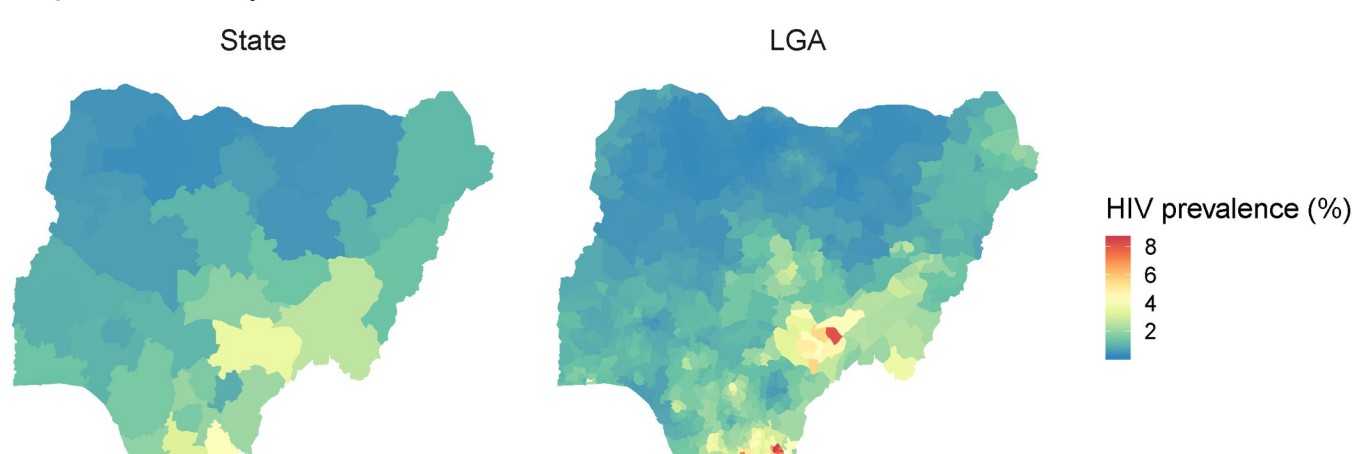

**Fig 1. Estimated HIV prevalence by state and local government area (n = 774) in Nigeria (2018).**

differences were large, especially at the ends of the distribution: the highest estimated prevalence (in Okobo, Akwa Ibom) was 37.7 times higher than the lowest (in Kankia, Katsina), and the 90th percentile was 6.4 times larger than the 10th percentile.

Differences in HIV prevalence between LGAs were greater than differences at the state level; whereas the maximum prevalence among states was 4.3% (95% UI: 3.7%–4.9%), it was nearly twice as high in the highest prevalence LGA (8.5% [95% UI: 5.8%–12.2%]). Both absolute and relative differences were therefore more extreme: whereas the maximum state-level prevalence estimate was 12.5 times larger than the minimum, the maximum LGA-level estimate was 37.7 times larger.

ART coverage ranged from 21.9% (95% UI: 9.9%–37.8%) in Yabo, Sokoto, the LGA with the lowest percentage of PLHIV receiving ART, to 81.9% (95% UI: 73.2%–88.5%) coverage in Tarka, Benue, the LGA with the highest percentage (Fig 2). The mean coverage was 45.0%,

## ART coverage by state and LGA

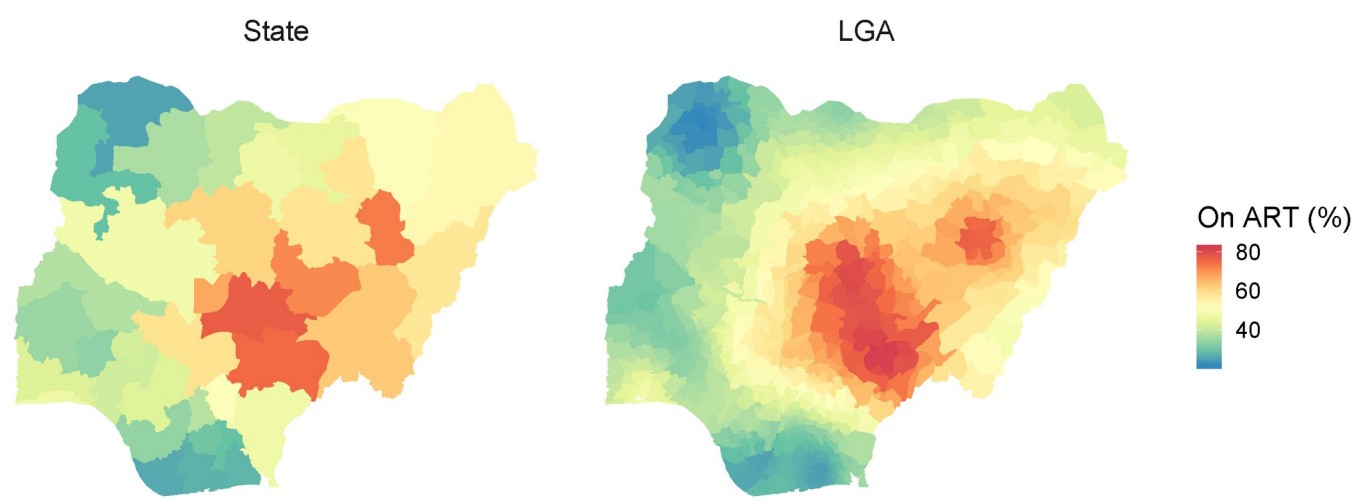

**Fig 2. Estimated antiretroviral therapy coverage by state and local government area (n = 774) in Nigeria (2018).**

with a 38.8 percentage point difference between the 10th and 90th percentiles. Although absolute differences were large, relative differences in coverage were smaller than differences in prevalence: the highest ART coverage estimate was 3.7 times larger than the lowest, and the 90th percentile was 2.4 times greater than the 10th. LGAs with higher ART coverage (>80%) were clustered in the Central and Eastern areas of Nigeria, with very low ART coverage (<30%) observed in LGAs in the South and Northwest regions.

Absolute and relative differences in ART coverage were also reflected in the LGA-level VLS rates: the lowest LGA VLS rate was 17.8% (Kolokuma and Opokuma [95% UI: 12.3%–24.6%]), compared to the highest at 71.0% (Tarka [95% UI: 62.5%–78.3%])—a 53.2 percentage point difference (Fig 3). The mean VLS rate was 36.2%, and there was an absolute difference of 34.3 percentage points between the 10th and 90th percentiles. Relative differences were also similar to differences in ART coverage.

### Differences between LGAs within states

The mean absolute difference between lowest and highest within-state LGA-level HIV prevalence was 1.7 percentage points and ranged from a 0.2 to a 7.2 percentage point difference (Table 3). Although there was a difference of less than 1 percentage point between the lowest and highest LGA estimate in 15 states, three states had a difference of more than 5 percentage points, including Rivers (5.4 percentage points), Akwa Ibom (6.2 percentage points), and Benue (7.2 percentage points). Relative differences were also substantial. Of the 36 states and FCT, 33 had a 2 times difference between the LGAs with the highest and lowest HIV prevalence estimates.

The mean absolute difference in ART coverage between LGAs within states was 16.7 percentage points and ranged from 3.5 to 38.5 percentage points (Table 4). Absolute differences tended to be large, with a difference of more than 10 percentage points between the highest and lowest coverage in 30 of 37 states (81%). In three states, the highest and the lowest LGA coverage differed by more than 30 percentage points: Niger (32.0 percentage points), Kaduna (32.0 percentage points), and Cross River (38.5 percentage points).

Viral suppression prevalence by state and LGA

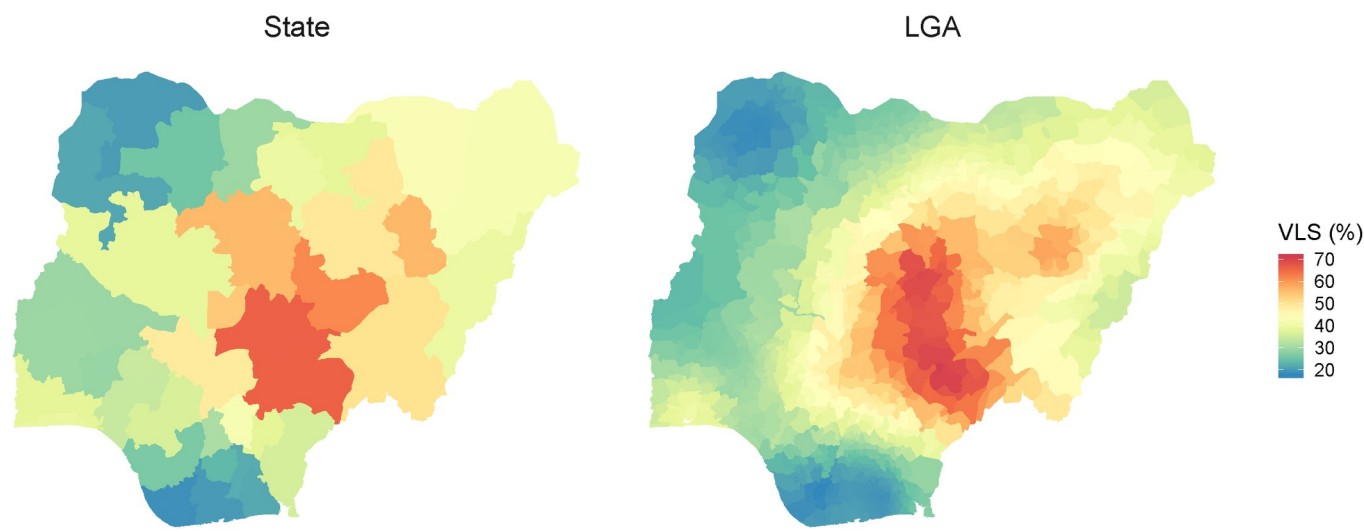

**Fig 3. Estimated viral load suppression rate by state and local government area (n = 774) in Nigeria (2018).**

**Table 3. Differences in in HIV prevalence between local government areas within states in Nigeria (2018).**

| State | Number of LGAs | HIV prev. (%) | 95% UI | Min. | Max. | Max.—Min. | Max./Min. | 10th perc. | 90th perc. | 90th - 10th | 90th/10th |
|---|---|---|---|---|---|---|---|---|---|---|---|
| Abia | 17 | 2.0 | 1.7–2.3 | 1.6 | 2.8 | 1.2 | 1.8 | 1.7 | 2.4 | 0.7 | 1.4 |
| Adamawa | 21 | 1.4 | 1.1–1.6 | 0.7 | 2.5 | 1.8 | 3.4 | 0.8 | 2.0 | 1.2 | 2.4 |
| Akwa Ibom | 31 | 4.3 | 3.7–4.9 | 2.4 | 8.5 | 6.2 | 3.6 | 3.1 | 7.6 | 4.4 | 2.4 |
| Anambra | 21 | 2.0 | 1.7–2.4 | 1.4 | 3.7 | 2.3 | 2.6 | 1.5 | 2.6 | 1.1 | 1.7 |
| Bauchi | 20 | 0.5 | 0.4–0.6 | 0.3 | 0.7 | 0.5 | 2.6 | 0.3 | 0.7 | 0.4 | 2.1 |
| Bayelsa | 8 | 1.9 | 1.5–2.3 | 1.4 | 2.8 | 1.4 | 2.0 | 1.5 | 2.8 | 1.3 | 1.9 |
| Benue | 23 | 3.6 | 3.1–4.2 | 1.0 | 8.1 | 7.2 | 8.5 | 1.2 | 5.4 | 4.3 | 4.7 |
| Borno | 27 | 1.1 | 0.9–1.4 | 0.7 | 2.0 | 1.3 | 2.8 | 0.8 | 1.7 | 0.9 | 2.2 |
| Cross River | 18 | 2.1 | 1.8–2.4 | 0.9 | 4.3 | 3.4 | 4.6 | 1.2 | 2.6 | 1.4 | 2.1 |
| Delta | 25 | 1.5 | 1.2–1.9 | 0.7 | 2.9 | 2.2 | 4.1 | 0.9 | 2.1 | 1.1 | 2.2 |
| Ebonyi | 13 | 0.9 | 0.7–1.1 | 0.6 | 1.6 | 1.1 | 2.9 | 0.6 | 1.5 | 0.9 | 2.5 |
| Edo | 18 | 1.5 | 1.3–1.8 | 0.7 | 2.5 | 1.8 | 3.5 | 1.1 | 1.8 | 0.7 | 1.6 |
| Ekiti | 16 | 0.8 | 0.6–1.1 | 0.4 | 1.4 | 1.0 | 3.3 | 0.5 | 1.0 | 0.5 | 2.0 |
| Enugu | 17 | 1.6 | 1.3–2.0 | 1.2 | 2.5 | 1.3 | 2.1 | 1.2 | 2.2 | 0.9 | 1.8 |
| FCT | 6 | 1.4 | 1.1–1.7 | 1.0 | 1.7 | 0.8 | 1.8 | 1.0 | 1.6 | 0.6 | 1.7 |
| Gombe | 11 | 1.0 | 0.8–1.2 | 0.4 | 2.6 | 2.2 | 6.6 | 0.5 | 2.1 | 1.7 | 4.8 |
| Imo | 27 | 1.6 | 1.3–1.9 | 0.8 | 2.3 | 1.5 | 2.8 | 1.0 | 2.0 | 1.0 | 2.0 |
| Jigawa | 27 | 0.4 | 0.3–0.6 | 0.3 | 0.7 | 0.4 | 2.5 | 0.3 | 0.6 | 0.2 | 1.7 |
| Kaduna | 23 | 1.0 | 0.8–1.2 | 0.4 | 3.0 | 2.6 | 7.0 | 0.5 | 2.2 | 1.7 | 4.4 |
| Kano | 44 | 0.6 | 0.5–0.9 | 0.3 | 0.9 | 0.6 | 3.0 | 0.4 | 0.8 | 0.3 | 1.8 |
| Katsina | 34 | 0.4 | 0.3–0.5 | 0.2 | 0.7 | 0.5 | 3.0 | 0.3 | 0.5 | 0.2 | 1.9 |
| Kebbi | 21 | 0.6 | 0.4–0.7 | 0.4 | 0.9 | 0.6 | 2.6 | 0.4 | 0.8 | 0.4 | 2.0 |
| Kogi | 21 | 1.1 | 0.9–1.3 | 0.8 | 2.0 | 1.2 | 2.6 | 0.9 | 1.2 | 0.3 | 1.4 |
| Kwara | 16 | 1.0 | 0.8–1.2 | 0.7 | 1.5 | 0.8 | 2.2 | 0.7 | 1.3 | 0.6 | 1.9 |
| Lagos | 20 | 1.3 | 1.1–1.6 | 0.8 | 2.2 | 1.4 | 2.7 | 1.0 | 1.6 | 0.6 | 1.6 |
| Nasarawa | 13 | 1.8 | 1.5–2.2 | 1.2 | 4.1 | 3.0 | 3.5 | 1.2 | 2.7 | 1.5 | 2.2 |
| Niger | 25 | 0.6 | 0.5–0.8 | 0.4 | 1.2 | 0.8 | 2.8 | 0.4 | 0.8 | 0.4 | 1.9 |
| Ogun | 20 | 1.4 | 1.1–1.7 | 0.8 | 1.8 | 1.0 | 2.3 | 0.9 | 1.5 | 0.6 | 1.7 |
| Ondo | 18 | 1.0 | 0.8–1.3 | 0.6 | 1.4 | 0.8 | 2.3 | 0.8 | 1.3 | 0.5 | 1.6 |
| Osun | 30 | 1.0 | 0.8–1.3 | 0.5 | 1.4 | 0.9 | 2.8 | 0.8 | 1.2 | 0.4 | 1.5 |
| Oyo | 33 | 0.9 | 0.7–1.2 | 0.7 | 1.2 | 0.5 | 1.6 | 0.8 | 1.1 | 0.3 | 1.4 |
| Plateau | 17 | 1.4 | 1.1–1.7 | 0.5 | 2.0 | 1.4 | 3.8 | 1.0 | 1.6 | 0.7 | 1.7 |
| Rivers | 23 | 3.2 | 2.8–3.7 | 1.9 | 7.4 | 5.4 | 3.8 | 2.1 | 5.1 | 3.0 | 2.4 |
| Sokoto | 23 | 0.5 | 0.3–0.6 | 0.3 | 0.9 | 0.6 | 3.0 | 0.3 | 0.6 | 0.3 | 1.9 |
| Taraba | 16 | 2.6 | 2.3–3.0 | 1.1 | 4.4 | 3.3 | 3.9 | 1.8 | 3.3 | 1.5 | 1.8 |
| Yobe | 17 | 0.5 | 0.4–0.6 | 0.3 | 1.2 | 0.8 | 3.5 | 0.3 | 0.7 | 0.4 | 2.2 |
| Zamfara | 14 | 0.34 | 0.25–0.45 | 0.3 | 0.5 | 0.2 | 1.8 | 0.3 | 0.4 | 0.1 | 1.5 |
| **National** | **774** | **1.36** | **1.30–1.44** | **0.2** | **8.5** | **8.3** | **37.7** | **0.4** | **2.6** | **2.2** | **6.4** |

Differences in the distribution of VLS within states were similar to differences in ART coverage. The mean absolute difference in VLS rates between LGAs within states was 14.3 percentage points, compared to a range of 2.0 to 34.5 percentage points (**Table 5**). Two states, Cross River and Kaduna, showed differences of greater than 30 percentage points between the LGAs with the lowest and the highest VLS prevalence among PLHIV. Similarly, two states, Niger and Cross River, contained LGAs with VLS rates more than double the prevalence in the LGA with the lowest rates of VLS.

**Table 4. Differences in antiretroviral therapy coverage among people living with HIV between local government areas within states in Nigeria (2018).**

| State (# of LGAs) | Number of LGAs | ART cov. (%) | 95% UI | Min. | Max. | Max.—Min. | Max./Min. | 10th perc. | 90th perc. | 90th - 10th | 90th/10th |
|---|---|---|---|---|---|---|---|---|---|---|---|
| Abia | 17 | 29.3 | 24.3–34.7 | 25.4 | 37.2 | 11.9 | 1.5 | 26.1 | 34.1 | 8.0 | 1.3 |
| Adamawa | 21 | 57.6 | 46.8–67.3 | 48.8 | 66.0 | 17.2 | 1.4 | 50.4 | 62.7 | 12.3 | 1.2 |
| Akwa Ibom | 31 | 27.9 | 23.4–32.9 | 24.7 | 33.9 | 9.2 | 1.4 | 25.2 | 32.5 | 7.3 | 1.3 |
| Anambra | 21 | 38.5 | 31.7–45.2 | 33.5 | 46.9 | 13.4 | 1.4 | 35.5 | 42.6 | 7.0 | 1.2 |
| Bauchi | 20 | 58.1 | 47.7–67.9 | 47.1 | 68.4 | 21.3 | 1.5 | 50.5 | 64.4 | 14.0 | 1.3 |
| Bayelsa | 8 | 26.3 | 19.5–34.4 | 25.2 | 28.7 | 3.5 | 1.1 | 25.4 | 28.2 | 2.9 | 1.1 |
| Benue | 23 | 75.0 | 69.7–79.7 | 58.6 | 81.9 | 23.3 | 1.4 | 63.8 | 81.5 | 17.6 | 1.3 |
| Borno | 27 | 53.6 | 36.2–69.2 | 42.7 | 71.0 | 28.3 | 1.7 | 44.7 | 64.3 | 19.6 | 1.4 |
| Cross River | 18 | 47.3 | 41.0–54.0 | 33.0 | 71.4 | 38.5 | 2.2 | 34.0 | 68.4 | 34.4 | 2.0 |
| Delta | 25 | 34.2 | 27.1–42.1 | 25.7 | 42.8 | 17.1 | 1.7 | 28.9 | 40.8 | 11.9 | 1.4 |
| Ebonyi | 13 | 47.4 | 38.6–56.0 | 38.2 | 54.0 | 15.8 | 1.4 | 38.9 | 53.0 | 14.2 | 1.4 |
| Edo | 18 | 43.6 | 35.8–51.3 | 37.5 | 53.6 | 16.1 | 1.4 | 38.7 | 51.4 | 12.6 | 1.3 |
| Ekiti | 16 | 42.1 | 31.8–53.1 | 36.6 | 51.7 | 15.1 | 1.4 | 38.0 | 47.2 | 9.1 | 1.2 |
| Enugu | 17 | 51.0 | 43.9–59.0 | 41.0 | 59.4 | 18.4 | 1.4 | 42.3 | 57.9 | 15.6 | 1.4 |
| FCT | 6 | 67.5 | 58.5–76.0 | 59.6 | 70.5 | 10.9 | 1.2 | 60.5 | 69.8 | 9.3 | 1.2 |
| Gombe | 11 | 72.8 | 63.5–80.4 | 63.7 | 77.8 | 14.1 | 1.2 | 66.9 | 76.6 | 9.7 | 1.1 |
| Imo | 27 | 30.5 | 24.7–36.6 | 27.9 | 33.8 | 5.9 | 1.2 | 29.0 | 32.2 | 3.2 | 1.1 |
| Jigawa | 27 | 45.4 | 33.1–57.9 | 36.1 | 56.3 | 20.2 | 1.6 | 38.0 | 51.0 | 13.0 | 1.3 |
| Kaduna | 23 | 61.0 | 50.4–71.4 | 48.1 | 80.1 | 32.0 | 1.7 | 48.9 | 79.5 | 30.6 | 1.6 |
| Kano | 44 | 46.9 | 34.4–61.0 | 40.0 | 57.8 | 17.8 | 1.4 | 42.7 | 50.5 | 7.8 | 1.2 |
| Katsina | 34 | 39.6 | 26.5–54.7 | 33.3 | 48.3 | 15.0 | 1.5 | 34.3 | 47.5 | 13.2 | 1.4 |
| Kebbi | 21 | 29.4 | 18.8–42.2 | 22.2 | 36.4 | 14.1 | 1.6 | 22.7 | 36.1 | 13.4 | 1.6 |
| Kogi | 21 | 58.7 | 49.7–67.2 | 47.7 | 68.7 | 21.0 | 1.4 | 51.2 | 64.2 | 13.0 | 1.3 |
| Kwara | 16 | 37.8 | 28.6–49.1 | 30.8 | 48.8 | 18.0 | 1.6 | 34.1 | 43.4 | 9.3 | 1.3 |
| Lagos | 20 | 46.8 | 39.1–55.1 | 42.1 | 48.2 | 6.1 | 1.1 | 45.1 | 48.0 | 2.9 | 1.1 |
| Nasarawa | 13 | 76.9 | 70.6–82.2 | 66.0 | 79.9 | 13.9 | 1.2 | 74.2 | 79.4 | 5.1 | 1.1 |
| Niger | 25 | 48.0 | 38.5–58.1 | 34.3 | 66.3 | 32.0 | 1.9 | 36.6 | 61.5 | 24.9 | 1.7 |
| Ogun | 20 | 43.6 | 36.4–51.1 | 37.2 | 47.5 | 10.3 | 1.3 | 39.0 | 46.1 | 7.1 | 1.2 |
| Ondo | 18 | 41.4 | 33.1–50.2 | 35.3 | 52.8 | 17.5 | 1.5 | 35.8 | 52.1 | 16.2 | 1.5 |
| Osun | 30 | 33.8 | 24.6–43.1 | 32.0 | 38.2 | 6.3 | 1.2 | 32.3 | 35.4 | 3.0 | 1.1 |
| Oyo | 33 | 34.9 | 25.9–45.3 | 30.0 | 39.7 | 9.7 | 1.3 | 30.9 | 37.9 | 6.9 | 1.2 |
| Plateau | 17 | 71.3 | 63.2–78.2 | 62.7 | 78.4 | 15.6 | 1.2 | 64.4 | 75.5 | 11.2 | 1.2 |
| Rivers | 23 | 27.3 | 21.7–33.2 | 25.2 | 29.4 | 4.2 | 1.2 | 25.9 | 28.4 | 2.6 | 1.1 |
| Sokoto | 23 | 25.6 | 13.9–40.6 | 21.9 | 34.5 | 12.7 | 1.6 | 22.4 | 29.9 | 7.4 | 1.3 |
| Taraba | 16 | 62.7 | 56.1–68.7 | 52.5 | 77.1 | 24.6 | 1.5 | 55.9 | 71.2 | 15.3 | 1.3 |
| Yobe | 17 | 50.7 | 35.8–67.0 | 39.5 | 65.4 | 25.9 | 1.7 | 40.8 | 60.3 | 19.4 | 1.5 |
| Zamfara | 14 | 37.1 | 21.7–55.0 | 25.1 | 45.1 | 20.1 | 1.8 | 29.6 | 42.2 | 12.6 | 1.4 |
| **National** | **774** | **45.3** | **42.8–48.0** | **21.9** | **81.9** | **60.1** | **3.7** | **28.0** | **66.8** | **38.8** | **2.4** |

## Discussion

Our study comprehensively quantify subnational HIV prevalence, ART coverage, and VLS in Nigeria in 2018. National and state-level estimates along the continuum of care provide important information for policymakers for allocating resources effectively to address the HIV epidemic. However, our findings highlight the advantages of sub-national estimates when geographically prioritizing interventions and refining existing HIV care and treatment programs. We found large absolute and relative differences in ART coverage and VLS between

**Table 5. Differences in viral load suppression among people living with HIV between local government areas within states in Nigeria (2018).**

| State (# of LGAs) | Number of LGAs | State VLS (%) | 95% UI | Min. | Max. | Max.—Min. | Max./Min. | 10th perc. | 90th perc. | 90th - 10th | 90th/10th |
|---|---|---|---|---|---|---|---|---|---|---|---|
| Abia | 17 | 22.0 | 18.2–26.2 | 19.2 | 29.6 | 10.3 | 1.5 | 19.5 | 26.7 | 7.2 | 1.4 |
| Adamawa | 21 | 39.5 | 31.5–48.4 | 34.9 | 48.6 | 13.7 | 1.4 | 36.3 | 45.7 | 9.4 | 1.3 |
| Akwa Ibom | 31 | 21.6 | 17.7–26.2 | 19.0 | 25.5 | 6.5 | 1.3 | 19.3 | 24.3 | 5.1 | 1.3 |
| Anambra | 21 | 31.2 | 25.6–37.1 | 25.9 | 39.9 | 13.9 | 1.5 | 28.5 | 35.3 | 6.8 | 1.2 |
| Bauchi | 20 | 49.3 | 39.3–59.2 | 37.5 | 56.4 | 19.0 | 1.5 | 40.7 | 56.1 | 15.4 | 1.4 |
| Bayelsa | 8 | 18.8 | 13.4–25.4 | 17.8 | 20.0 | 2.2 | 1.1 | 17.9 | 19.8 | 1.9 | 1.1 |
| Benue | 23 | 66.0 | 60.6–70.9 | 50.5 | 71.0 | 20.6 | 1.4 | 53.7 | 70.1 | 16.4 | 1.3 |
| Borno | 27 | 42.4 | 28.0–58.8 | 35.9 | 54.9 | 19.0 | 1.5 | 37.8 | 49.9 | 12.1 | 1.3 |
| Cross River | 18 | 35.7 | 29.6–42.3 | 25.2 | 59.6 | 34.5 | 2.4 | 25.7 | 57.8 | 32.1 | 2.3 |
| Delta | 25 | 26.2 | 20.4–32.6 | 18.4 | 35.2 | 16.9 | 1.9 | 20.5 | 32.5 | 12.0 | 1.6 |
| Ebonyi | 13 | 37.7 | 30.9–45.3 | 30.1 | 45.9 | 15.9 | 1.5 | 30.7 | 44.7 | 14.0 | 1.5 |
| Edo | 18 | 36.3 | 29.9–43.5 | 28.8 | 46.6 | 17.9 | 1.6 | 31.3 | 44.7 | 13.4 | 1.4 |
| Ekiti | 16 | 34.4 | 25.6–44.0 | 30.7 | 40.1 | 9.4 | 1.3 | 31.8 | 37.1 | 5.3 | 1.2 |
| Enugu | 17 | 42.1 | 35.0–48.9 | 33.6 | 49.4 | 15.9 | 1.5 | 34.3 | 48.5 | 14.2 | 1.4 |
| FCT | 6 | 54.1 | 45.8–62.6 | 47.5 | 57.1 | 9.6 | 1.2 | 48.3 | 56.3 | 8.0 | 1.2 |
| Gombe | 11 | 56.4 | 48.1–64.6 | 50.7 | 59.5 | 8.8 | 1.2 | 51.7 | 58.6 | 6.9 | 1.1 |
| Imo | 27 | 23.4 | 19.0–28.9 | 20.5 | 26.5 | 5.9 | 1.3 | 21.4 | 25.2 | 3.8 | 1.2 |
| Jigawa | 27 | 38.2 | 27.0–49.4 | 29.3 | 46.2 | 16.9 | 1.6 | 31.0 | 43.8 | 12.8 | 1.4 |
| Kaduna | 23 | 56.2 | 47.7–64.7 | 35.3 | 69.6 | 34.3 | 2.0 | 39.5 | 69.0 | 29.5 | 1.7 |
| Kano | 44 | 39.5 | 27.8–51.8 | 32.0 | 52.1 | 20.1 | 1.6 | 33.7 | 44.8 | 11.1 | 1.3 |
| Katsina | 34 | 29.6 | 18.9–43.7 | 25.9 | 36.8 | 10.9 | 1.4 | 26.5 | 34.4 | 7.9 | 1.3 |
| Kebbi | 21 | 21.4 | 11.9–33.1 | 18.5 | 26.9 | 8.5 | 1.5 | 18.7 | 25.5 | 6.8 | 1.4 |
| Kogi | 21 | 48.8 | 41.0–57.2 | 36.0 | 58.9 | 22.9 | 1.6 | 40.4 | 54.6 | 14.2 | 1.4 |
| Kwara | 16 | 29.4 | 21.4–38.6 | 23.6 | 36.5 | 12.9 | 1.5 | 26.7 | 33.9 | 7.3 | 1.3 |
| Lagos | 20 | 39.1 | 32.1–47.0 | 36.0 | 40.4 | 4.4 | 1.1 | 38.1 | 39.6 | 1.4 | 1.0 |
| Nasarawa | 13 | 66.2 | 59.6–72.4 | 55.6 | 68.6 | 13.1 | 1.2 | 63.3 | 68.5 | 5.2 | 1.1 |
| Niger | 25 | 38.4 | 30.3–47.8 | 25.0 | 53.1 | 28.1 | 2.1 | 26.4 | 48.4 | 22.0 | 1.8 |
| Ogun | 20 | 37.8 | 30.8–45.5 | 30.7 | 40.4 | 9.6 | 1.3 | 33.7 | 39.3 | 5.7 | 1.2 |
| Ondo | 18 | 34.2 | 26.7–42.1 | 30.1 | 42.4 | 12.3 | 1.4 | 30.7 | 40.9 | 10.2 | 1.3 |
| Osun | 30 | 29.1 | 21.3–37.5 | 27.4 | 31.4 | 4.0 | 1.1 | 27.7 | 30.3 | 2.6 | 1.1 |
| Oyo | 33 | 29.7 | 21.7–38.8 | 24.0 | 34.7 | 10.8 | 1.4 | 24.5 | 33.1 | 8.6 | 1.4 |
| Plateau | 17 | 61.6 | 53.2–69.2 | 47.6 | 69.2 | 21.6 | 1.5 | 50.2 | 66.8 | 16.6 | 1.3 |
| Rivers | 23 | 19.9 | 15.6–24.7 | 18.6 | 20.5 | 2.0 | 1.1 | 19.3 | 20.5 | 1.1 | 1.1 |
| Sokoto | 23 | 20.2 | 9.9–33.6 | 18.3 | 23.5 | 5.2 | 1.3 | 18.5 | 22.2 | 3.7 | 1.2 |
| Taraba | 16 | 50.3 | 44.0–56.4 | 40.1 | 61.8 | 21.7 | 1.5 | 41.2 | 59.3 | 18.1 | 1.4 |
| Yobe | 17 | 42.7 | 28.6–57.8 | 32.8 | 51.8 | 19.0 | 1.6 | 33.5 | 48.8 | 15.2 | 1.5 |
| Zamfara | 14 | 25.5 | 14.3–40.1 | 19.9 | 30.2 | 10.2 | 1.5 | 21.8 | 28.9 | 7.1 | 1.3 |
| **National** | **774** | **36.6** | **34.8–38.6** | **17.8** | **71.0** | **53.2** | **4.0** | **20.5** | **54.8** | **34.3** | **2.7** |

LGAs, both within and between states. Further, we found that there was a difference of more than 10 percentage points between the highest and the lowest performing LGAs in 30 states, indicating dramatic variation in treatment access. We observed a similar pattern in our analysis of VLS rates. This range in performance is estimated to be even larger in some states, with differences in ART coverage and VLS >30 percentage points in some areas.

Reliance solely on state-level estimates may therefore be misleading in identifying high-priority locations for program interventions. Consistent with other studies demonstrating localized HIV epidemics [26], our results show LGAs with high HIV prevalence in both high- and

low-prevalence states. In the states of Gombe and Kaduna, where HIV prevalence (1.0%) is less than the national average, we found LGAs where prevalence approached or surpassed twice the national average. Similar variability existed when examining the health system response through ART coverage and VLS, where high and low rates of effective coverage at the state level masked important variation that identified poor performance in otherwise high-performing states and vice versa. Interestingly, we saw relatively little variability in LGA performance between ART coverage and VLS.

In response to the detailed data obtained through NAIIS, the Government of Nigeria adopted the Nigeria ART Surge Plan in March of 2019. This plan called for increasing the number of PLHIV on ART by 500,000 by September 2020 [19]. To achieve this objective, the government, in conjunction with international partners, intensified efforts for case finding, treatment initiation, and retention in 10 priority states. As part of the planning process, the government decided a standardized approach with centralized control was unlikely to yield results quickly enough to reach the Surge Plan objectives. Therefore, the Government of Nigeria decentralized decision making and also increased accountability. As part of this process, local policymakers used LGA estimates when prioritizing where to perform community-based testing and improve facility-based treatment.

The implementation planning process in Nigeria also provided an example of how to gain stakeholder buy-in from the onset of the geospatial analysis to ensure utilization of results. Including policymakers and implementing partners in the estimation phase of this work provided sufficient background for these stakeholders to have confidence in the geospatial estimation results. By consistently including key local partners throughout the estimation process, the Government of Nigeria was able to ensure that LGA-level estimates were used in local intervention planning. This process of inclusion has been described as being critical in other locations [27]. Continued triangulation of data and monitoring of progress toward goals set for areas identified through small area estimates can help ensure emerging areas of need are identified.

At the same time, geospatial estimates cannot be the sole determinant of geographically targeted interventions. In particular, including equity as a key component of the decision-making process is critical. The Global Fund, the US President's Emergency Plan for AIDS Relief, and other international donors have provided guidance on the importance of ensuring equitable access to treatment for vulnerable populations, including young women and girls, sex workers, people who inject drugs, and other marginalized populations [28, 29]. Furthermore, geographically targeted interventions are only as successful as the health system allows. One study found that a distribution strategy focusing exclusively on high prevalence districts was less effective than a strategy combining geographic targeting with a minimum package of widely distributed cost-effective interventions; incorporating geographic targeting in the context of a basic package of national interventions could improve outcomes [6]. An effective model is adaptable and uses geospatial data to identify geographic areas for programmatic prioritization and uses program data to assess targeted interventions continuously.

Our study has several limitations. Most importantly, the accuracy of our estimates depends upon the quality of underlying data. NAIIS was one of the largest HIV population-based surveys ever conducted [20, 30]; nonetheless, important gaps in survey coverage occurred. These gaps include non-coverage of some enumeration areas in North-Eastern Nigeria due to security challenges and enumeration areas in Southern Nigeria due to flooding. As is the case for all household surveys, NAIIS was also impacted by non-response. Although household, individual, and blood draw response rates were high and attempts were made to weight the data to mitigate the impact of non-response, it is possible that some non-response bias remains.

To protect respondent confidentiality, GPS coordinates associated with each survey cluster were randomly displaced, which introduces another potential source of error into our analysis. Our modeling strategy relied on spatial smoothing to generate stable estimates despite small sample sizes in most locations. In cases where the true underlying surface had discontinuities, our model is likely to produce overly smooth estimates, which may in turn lead to an underestimation of geographic inequalities.

Differences in model specification can lead to meaningful differences in the resulting estimates of HIV prevalence, ART coverage, and VLS. While we undertook a cross-validation exercise to select our model among several potential candidate models, cross-validation is inherently limited in this context since the number of observations in any individual cluster is small, and the observed prevalence in that small sample may differ substantially from true underlying prevalence, making it difficult to assess model performance confidently in locations where data were withheld.

Finally, our estimates of PLHIV and our aggregated (LGA-, state-, and national-level) estimates of all indicators incorporated population estimates from WorldPop. These estimates are associated with some uncertainty, which we were unable to propagate through our modeling process and is therefore not captured by the uncertainty intervals reported. Moreover, any errors in the WorldPop estimates could impact the estimates reported here, particularly the estimates of the number of PLHIV.

## Conclusion

Despite considerable progress in expanding access to HIV testing and ART, geographic disparities in HIV prevalence, ART coverage, and VLS persist at the state and sub-state levels. Our localized estimates of these key indicators will position the government, donors, and other implementing partners to conduct targeted implementation planning ("microplanning") and effectively prioritize resource distribution to achieve epidemic control with greater efficiency. Our results show substantial variation between LGA for all three indicators, both within and between states. As expected, states with large differences in ART coverage between LGAs also showed large differences in VLS, and these differences were present in states with both high and low levels of effective ART coverage—indicating that state-level geographic targeting may be insufficient to address coverage gaps. In addition, including both local and national stakeholders throughout the estimation process allowed for increased use of the estimates to inform implementation planning. When combined with supplementary data on key populations and stakeholder inclusion, these estimates can be used to identify high priority areas effectively and distribute resources equitably and efficiently, with the potential to accelerate progress toward HIV epidemic control in Nigeria.

## Acknowledgments

This work was undertaken in collaboration with the U.S. Centers for Disease Control and Prevention, the Federal Ministry of Health, the National Agency for the Control of AIDS (NACA), and the University of Maryland Baltimore (UMB) as part of the Nigeria HIV/AIDS Indicator and Impact Survey (NAIIS). The findings are those of the authors and do not necessarily represent the official position of the government agencies listed.

The NAIIS Group includes Principal Investigators: Isaac Adewole (Federal Ministry of Health), Sani Aliyu (National Agency for the Control of AIDS), Mahesh Swaminathan (CDC Nigeria), Megan Bronson (CDC Atlanta), Manhattan Charurat (University of Maryland, Baltimore); Co-Investigators: Evelyn Ngige, Sunday Aboje, Charles Nzelu, Emanuel Meribole,

Chike Ihekweazu, Chukuma Anyaike, Kayode Ogungbemi, Mukhtar Muhammad, Gregory Ashefor, Ibrahim Dalhatu, Ibrahim Jahun, Victor Sebastian, Ahmed Mukhtar, Tapdiyel Jelpe, Orji Bassey, McPaul Okoye, Aminu Yakubu, Bharat Parekh, Hetal Patel, Andrew Voetsch, Daniel B. Williams, Kristin Brown, Stephen McCracken, Anne McIntyre, Nibretie Workneh, Bryan Morris, Rex Gadama Mpazanje, Wondimagegnehu Alemu, Erasmus Morah, Gatien Ekanmian, Gambo Aliyu, Alash'le Abimiku, Bola Gobir, Mercy Niyang, Isiramen Olajide, Baffa Ibrahim, Stephen Ohakanu, Ryan Leo, Geoffrey Greenwell, Adedayo Adeyemi, Bamgboye Afolabi, Ekanem, Mustapha Jamda, Annie Chen, Otse Ogorry, Aminu Suleiman, Kolapo Usman, Ojor R. Ayemoba, Adebobola Bashorun; Collaborating Institutions: Federal Ministry of Health (FMOH), National Agency for the Control of AIDS (NACA), National Population Commission (NPopC), National Bureau of Statistics (NBS), the U.S. Centers for Disease Control and Prevention (CDC) Nigeria, CDC Atlanta, The Global Funds to Fight AIDS, Tuberculosis, and Malaria, University of Maryland Baltimore (UMB), ICF International, African Field Epidemiology Network, University of Washington, the Joint United Nations Programme on HIV and AIDS (UNAIDS), the World Health Organization (WHO), and the United Nations Children's Fund (UNICEF).

## Author Contributions

**Conceptualization:** Laura DWYER-LINDGREN, Man CHARURAT.

**Data curation:** Rukevwe ALIOGO, Akipu EHOCHE.

**Formal analysis:** Caitlin O'BRIEN-CARELLI, Krista STEUBEN, Casey K. JOHANNS, Laura DWYER-LINDGREN.

**Funding acquisition:** Man CHARURAT.

**Investigation:** Matthias ALAGI, Gambo ALIYU, Sani ALIYU, Man CHARURAT.

**Methodology:** Rukevwe ALIOGO, Akipu EHOCHE, Herbert C. DUBER.

**Project administration:** Matthias ALAGI, Jahun IBRAHIM, Stacie GREBY, Dalhatu IBRAHIM, Megan BRONSON, Man CHARURAT.

**Resources:** Emilio DIRLIKOV.

**Supervision:** Jahun IBRAHIM, Dalhatu IBRAHIM, Megan BRONSON, Gambo ALIYU, Mahesh SWAMINATHAN, Man CHARURAT.

**Writing – original draft:** Caitlin O'BRIEN-CARELLI, Krista STEUBEN.

**Writing – review & editing:** Kristen A. STAFFORD, Matthias ALAGI, Casey K. JOHANNS, Jahun IBRAHIM, Ray SHIRAISHI, Emilio DIRLIKOV, Dalhatu IBRAHIM, Megan BRONSON, Gambo ALIYU, Sani ALIYU, Laura DWYER-LINDGREN, Mahesh SWAMINATHAN, Herbert C. DUBER, Man CHARURAT.

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
