## [Decision Letter · Decision Letter 0]

25 Jan 2022

PONE-D-21-31941Mapping HIV prevalence in Nigeria using small area estimates to develop a targeted HIV intervention strategyPLOS ONE

Dear Dr. Charurat,

Thank you for submitting your manuscript to PLOS ONE. After careful consideration, we feel that it has merit but does not fully meet PLOS ONE’s publication criteria as it currently stands. Therefore, we invite you to submit a revised version of the manuscript that addresses the points raised during the review process.

We look forward to receiving your revised manuscript.

Kind regards,

Clement Ameh Yaro, Ph.D

Academic Editor

PLOS ONE

Journal Requirements:

3. We note that Figure(s) 1, 2 and 3 in your submission contain [map/satellite] images which may be copyrighted. All PLOS content is published under the Creative Commons Attribution License (CC BY 4.0), which means that the manuscript, images, and Supporting Information files will be freely available online, and any third party is permitted to access, download, copy, distribute, and use these materials in any way, even commercially, with proper attribution. For these reasons, we cannot publish previously copyrighted maps or satellite images created using proprietary data, such as Google software (Google Maps, Street View, and Earth). For more information, see our copyright guidelines: http://journals.plos.org/plosone/s/licenses-and-copyright.

1. You may seek permission from the original copyright holder of Figure(s) 1, 2 and 3  to publish the content specifically under the CC BY 4.0 license.  

4. Please provide additional details regarding participant consent. In the ethics statement in the Methods and online submission information, please ensure that you have specified (1) whether consent was informed and (2) what type you obtained (for instance, written or verbal, and if verbal, how it was documented and witnessed). If your study included minors, state whether you obtained consent from parents or guardians. If the need for consent was waived by the ethics committee, please include this information.

Additional Editor Comments (if provided):

Dear Authors,

The following comments were raised by the reviewers. Kindly respond to these comments.

REVIEWER 1

Great efforts put in to utilized population-based HIV survey for evidence or targeted interventions to prevent new HIV infection and HIV control in a resource-limited nation is good innovation.

The STATISTICAL METHODS is robust and results unambiguous for policy guidance in HIV/AIDS programming. However, there are many claims that ensemble models used in your statistical analysis could be shrouded with generalization errors, how did you overcome overfitting models in your analysis?

Data access was adequate and useful in buttressing the credibility of the manuscript.

REVIEWER 2

The strength of the study lies on the used household surveys and GPS to analyze data for HIV prevalence, ART and VLS.

This is a better model than the State based approaches used presently in the country for policy making. However, the challenge is that the use of these databases have underlying assumptions, as stated by the authors as limitations to the study, use of household data may not correctly capture high risk groups, vulnerable populations and marginalized communities. Also, the use of GPS may incorrectly capture coordinates for all the locations. These limitations will hamper the reliability of the results.

Suggestion: There are other software used by medical geographers and geospatial analysts that could be used alongside the data used in this study to further enhance the accuracy of the results and also eliminate the bias further. It is not out of place if the authors engage these specialists in order to further improve the quality of the work.

Specific suggestions: Replace "prevalence of ART" with "ART coverage"

Replace "prevalence of VL suppression" with "virus suppression rate"

Reviewers' comments:

Reviewer's Responses to Questions

**Comments to the Author**

1. Is the manuscript technically sound, and do the data support the conclusions?

Reviewer #1: Yes

Reviewer #2: Partly

2. Has the statistical analysis been performed appropriately and rigorously? 

Reviewer #1: Yes

Reviewer #2: Yes

3. Have the authors made all data underlying the findings in their manuscript fully available?

Reviewer #1: Yes

Reviewer #2: Yes

4. Is the manuscript presented in an intelligible fashion and written in standard English?

Reviewer #1: Yes

Reviewer #2: Yes

5. Review Comments to the Author

Reviewer #1: Great efforts put in to utilized population-based HIV survey for evidence or targeted interventions to prevent new HIV infection and HIV control in a resource-limited nation is good innovation. The STATISTICAL METHODS is robust and results unambiguous for policy guidance in HIV/AIDS programming. However, there are many claims that ensemble models used in your statistical analysis could be shrouded with generalization errors, how did you overcome overfitting models in your analysis?

Data access was adequate and useful in buttressing the credibility of the manuscript.

Reviewer #2: The strength of the study lies on the used household surveys and GPS to analyze data for HIV prevalence, ART and VLS. This is a better model than the State based approaches used presently in the country for policy making.

However, the challenge is that the use of these databases have underlying assumptions, as stated by the authors as limitations to the study, use of household data may not correctly capture high risk groups, vulnerable populations and marginalized communities. Also, the use of GPS may incorrectly capture coordinates for all the locations.

These limitations will hamper the reliability of the results.

Suggestion: There are other software used by medical geographers and geospatial analysts that could be used alongside the data used in this study to further enhance the accuracy of the results and also eliminate the bias further. It is not out of place if the authors engage these specialists in order to further improve the quality of the work.

Specific suggestions:

Replace "prevalence of ART" with "ART coverage"

Replace "prevalence of VL suppression" with "virus suppression rate"

6. PLOS authors have the option to publish the peer review history of their article (what does this mean?). If published, this will include your full peer review and any attached files.

Reviewer #1: No

Reviewer #2: No

---

## [Author Response · Author response to Decision Letter 0]

26 Mar 2022

REVIEWER 1

Great efforts put in to utilized population-based HIV survey for evidence or targeted interventions to prevent new HIV infection and HIV control in a resource-limited nation is good innovation.

The STATISTICAL METHODS is robust and results unambiguous for policy guidance in HIV/AIDS programming. However, there are many claims that ensemble models used in your statistical analysis could be shrouded with generalization errors, how did you overcome overfitting models in your analysis?

Data access was adequate and useful in buttressing the credibility of the manuscript.

Response: Thank you to the reviewer for their thorough review and insightful comments. We agree with the reviewer that there can be concerns about over-fitting with the type of ensemble model that we utilized for HIV prevalence (note that the models for ART and VLS do not include this ensemble modeling component). Following Bhatt and colleagues (citation 25 in the main text), as well as previous applications of a similar approach to modeling HIV prevalence (citation 1 in the main text), we mitigate the risk of over-fitting via a cross-validation approach. Specifically, the observations in the full dataset are divided into five groups and each of the sub-models is fit five times, excluding each group of data in turn. Out-of-sample predictions are then produced for the excluded data for each model, which leads to a full set of out-of-sample predictions for each sub-model. These out-of-sample predictions from the sub-models are then used in fitting the geo-statistical model. The sub-models are also fit on the full dataset, and this is used to generate a full set of in-sample predictions for each sub-model. These in-sample predictions, plus the fitted geo-statistical model, are then used to produce the final estimates. By leveraging out-of-sample predictions throughout the model fitting process, we prioritize models that generalize well, rather than models that potentially over-fit the observed data. 

REVIEWER 2

The strength of the study lies on the used household surveys and GPS to analyze data for HIV prevalence, ART and VLS.

This is a better model than the State based approaches used presently in the country for policy making. However, the challenge is that the use of these databases have underlying assumptions, as stated by the authors as limitations to the study, use of household data may not correctly capture high risk groups, vulnerable populations and marginalized communities. Also, the use of GPS may incorrectly capture coordinates for all the locations. These limitations will hamper the reliability of the results.

Suggestion: There are other software used by medical geographers and geospatial analysts that could be used alongside the data used in this study to further enhance the accuracy of the results and also eliminate the bias further. It is not out of place if the authors engage these specialists in order to further improve the quality of the work.

Response: Thank you to the reviewer for their careful review and thoughtful comments. We agree with the reviewer that a more detailed spatial analysis like this one reveals important additional information and enables insights beyond what is afforded by state-level analysis alone. We also agree that there are a number of challenges inherent in the analysis of household survey data in general and particularly in the context of geospatial analysis, including concerns related to survey coverage and representativeness and the errors introduced by displacing GPS coordinates in order to protect respondents’ confidentiality, and so we explicitly acknowledge these concerns in the section on limitations in the discussion. While there are a variety of approaches that have been employed for this type of detailed spatial analysis, we believe that the particular approach used in this manuscript is both appropriate given the research questions and comparable to methodologies regularly utilized by other experts in the field. We also note that a number of authors of this manuscript, including those most closely involved in the development of the statistical models, could reasonably be considered specialists in geospatial analysis. 

Specific suggestions: Replace "prevalence of ART" with "ART coverage"

Replace "prevalence of VL suppression" with "virus suppression rate"

Response: The revised manuscript includes the reviewers suggested changes.

---

## [Decision Letter · Decision Letter 1]

11 May 2022

Mapping HIV prevalence in Nigeria using small area estimates to develop a targeted HIV intervention strategy

PONE-D-21-31941R1

Dear Dr. Charurat,

We’re pleased to inform you that your manuscript has been judged scientifically suitable for publication and will be formally accepted for publication once it meets all outstanding technical requirements.

Kind regards,

Clement Ameh Yaro, Ph.D

Academic Editor

PLOS ONE

Additional Editor Comments (optional):

Dear Authors

Kindly effect the following minor corrections in the abstract as suggested by the second reviewer;

• Prevalence of ART still used in line 3 under design. Please replace with ART coverage

• Same with line 4, replace prevalence of VLS with VLS rate.

Reviewers' comments:

Reviewer's Responses to Questions

**Comments to the Author**

1. If the authors have adequately addressed your comments raised in a previous round of review and you feel that this manuscript is now acceptable for publication, you may indicate that here to bypass the “Comments to the Author” section, enter your conflict of interest statement in the “Confidential to Editor” section, and submit your "Accept" recommendation.

Reviewer #1: All comments have been addressed

Reviewer #2: All comments have been addressed

2. Is the manuscript technically sound, and do the data support the conclusions?

Reviewer #1: Yes

Reviewer #2: Yes

3. Has the statistical analysis been performed appropriately and rigorously? 

Reviewer #1: Yes

Reviewer #2: Yes

4. Have the authors made all data underlying the findings in their manuscript fully available?

Reviewer #1: Yes

Reviewer #2: Yes

5. Is the manuscript presented in an intelligible fashion and written in standard English?

Reviewer #1: Yes

Reviewer #2: Yes

6. Review Comments to the Author

Reviewer #1: Good paper for the scientific community to put to use. Th author has satisfactorily addressed the areas of concern earlier raised in the first submission and I am confident that it is good for publication.

Reviewer #2: COMMENTS:

Abstract:

• Prevalence of ART still used in line 3 under design. Please replace with ART coverage

• Same with line 4, replace prevalence of VLS with VLS rate.

7. PLOS authors have the option to publish the peer review history of their article (what does this mean?). If published, this will include your full peer review and any attached files.

Reviewer #1: **Yes: **Francis A Magaji

Reviewer #2: No

---

## [Editor Report · Acceptance letter]

26 May 2022

PONE-D-21-31941R1 

Mapping HIV prevalence in Nigeria using small area estimates to develop a targeted HIV intervention strategy 

Dear Dr. Charurat:

I'm pleased to inform you that your manuscript has been deemed suitable for publication in PLOS ONE. Congratulations! Your manuscript is now with our production department. 

Kind regards, 

on behalf of

Dr. Clement Ameh Yaro 

Academic Editor

PLOS ONE